# La(Ca)CrO_3_-Filled SiCN Precursor Thin Film Temperature Sensor Capable to Measure up to 1100 °C High Temperature

**DOI:** 10.3390/mi14091719

**Published:** 2023-08-31

**Authors:** Gonghan He, Yingping He, Lida Xu, Lanlan Li, Lingyun Wang, Zhenyin Hai, Daoheng Sun

**Affiliations:** Department of Mechanical and Electrical Engineering, Xiamen University, Xiamen 361005, China; 18750605740@163.com (Y.H.); superldxu@gmail.com (L.X.); lilanlan1006@163.com (L.L.); wangly@xmu.edu.cn (L.W.); haizhenyin@xmu.edu.cn (Z.H.)

**Keywords:** La(Ca)CrO_3_, SiCN precursor, thin film, temperature sensors, high temperature sensors

## Abstract

Thin-film sensors are regarded as advanced technologies for in situ condition monitoring of components operating in harsh environments, such as aerospace engines. Nevertheless, these sensors encounter challenges due to the high-temperature oxidation of materials and intricate manufacturing processes. This paper presents a simple method to fabricate high temperature-resistant oxidized SiCN precursor and La(Ca)CrO_3_ composite thin film temperature sensors by screen printing and air annealing. The developed sensor demonstrates a broad temperature response ranging from 200 °C to 1100 °C with negative temperature coefficients (NTC). It exhibits exceptional resistance to high-temperature oxidation and maintains performance stability. Notably, the sensor’s resistance changes by 3% after exposure to an 1100 °C air environment for 1 h. This oxidation resistance improvement surpasses the currently reported SiCN precursor thin-film sensors. Additionally, the sensor’s temperature coefficient of resistance (TCR) can reach up to −7900 ppm/°C at 200 °C. This strategy is expected to be used for other high-temperature thin-film sensors such as strain gauges, heat flux sensors, and thermocouples. There is great potential for applications in high-temperature field monitoring.

## 1. Introduction

The optimization and intelligent development of aeroengines require the participation of numerous types of sensors. Most of its core components operate in harsh environments such as high temperatures [1] (>1000 °C). Therefore, temperature control and monitoring are vital for them. Currently, conventional wire and MEMS block sensors struggle to withstand in the harsh environment of high temperature, high pressure, and high speed. In comparison, the sensor affects the flow pattern of the gas, causing interference with the vibrational modes of critical components [2]. Thin-film sensors have the advantages of fast response, nearly negligible interference with the component flow field, and integration on the component’s surface, making thin-film sensors one of the vital development directions for future harsh environment sensing [1,2,3,4,5,6,7,8,9,10,11].

However, there are challenges in fabricating thin-film sensors, both in manufacturing process methods and sensitive materials. On the one hand, conventional thin film manufacturing processes are expensive in terms of equipment, the need for limited part sizes, and complex fabrication processes, such as physical vapor deposition, chemical vapor deposition, and laser deposition [12,13,14,15]. For small-volume customization, the methods are more expensive to manufacture and slower to optimize iterations. On the other hand, thin-film sensors made of alloy materials struggle to withstand the long-term effects of high-temperature atmospheres with oxygen above 1000 °C [16,17,18,19]. Although NASA reported [20,21] that PdCr-based thin film sensors can reach 1100 °C, the cost is high, and the process is complicated. Therefore, they are not suitable for large-scale applications.

Screen printing technology offers several advantages, including its simplicity, scalability, and environmentally friendly processes that facilitate rapid manufacturing. Furthermore, the translucency of the screen mask allows for precise alignment [22,23]. Thus, the technology shows great potential for producing high-volume electronic thin film devices at a meager cost. When using screen printing technology to fabricate high-temperature thin-film sensors, the functional slurries must first develop to adapt to high-temperature environments. Polymer-derived ceramic (PDC) with excellent high-temperature resistance and liquid phase at room temperature is the ideal sensitive material for high-temperature sensors [4,10,24]. Cui et al. [24] pioneered the fabrication of high-temperature thin-film resistive temperature detectors (RTD) [25]. However, PDC thin-film sensors suffer serious high-temperature oxidation problems [5,9]. On the one hand, the free carbon in PDC undergoes oxidation above 400 °C [4], and on the other hand, the free silicon in PDC is affected by oxidation to form SiO_2_ [26]. To address this challenge of PDC thin-film sensors, Cui et al. proposed PDC antioxidant coating [9] or pyrolysis in vacuum [10] to enhance the antioxidant performance of PDC thin-film sensors. Chao et al. [7]. enhanced the antioxidant performance of PDC thin-film sensors by self-generation of the antioxidant layer by TiB_2_ filler. Although these methods have improved the oxidation resistance of the sensor, PDC thin-film sensors can only be used in air below 800 °C. However, meeting the requirements for aero-engine applications is a challenging task. The critical high-temperature monitoring range of 800 to 1100 °C is indispensable to cater to the developmental and measurement demands of aeroengines [27,28]. This importance is further underscored by the fact that the majority of turbine metallic alloys exhibit a temperature tolerance of about 950 °C [28].

To overcome the high-temperature oxidation challenge of sensors, NASA [29] reported that devices made of conductive oxides might be ideal for temperature measurements in harsh environments such as air/oxygen. Calcium lanthanum chromite La(Ca)CrO_3_ (LCC) is built upon the perovskite structure and characterized by an ABX3 crystal lattice arrangement, where A-site represents rare earth elements (such as La, Ca), and B-site involves transition metal elements (like Cr) [30,31,32]. This structural configuration can lead to defect sites and oxygen vacancies within the lattice, consequently influencing its conductive properties. La(Ca)CrO_3_ demonstrates strong oxygen ion conductivity at elevated temperatures due to the mobility of oxygen ions within the lattice. Apart from oxygen ion conduction, La(Ca)CrO_3_ also possesses electron conductivity. Khuong P. Ong et al. employed a combined approach of molecular modeling and experimental methods to enhance the conductivity of LaCrO_3_ by screening all possible partial substitutions of La with Group IIa elements. The researchers calculated the electronic structures of doped and undoped LaCrO_3_ and demonstrated the semiconductor state of m^2+^-doped LaCrO_3_. Theoretical and experimental results both indicate that doping with Ca, Sr, and Ba increases the conductivity of LaCrO_3_, with the Ca-doped variant achieving the highest conductivity [30]. Therefore, LCC is an excellent high-temperature sensing material. It is easy to think of combining the merits of PDC and LCC to make a slurry. Compared to sensors fabricated using other materials filled SiCN, the sensor produced from the LCC-filled polymer-derived SiCN (LCC-SiCN) composite is introduced for the first time. This novel sensor exhibits a higher temperature tolerance and enhanced oxidation resistance. The innovative LCC-SiCN composite sensor is anticipated to overcome the limitations faced by current thin film sensors, which struggle to withstand oxidation at ultra-high temperatures. This advancement holds significant practical value in various applications.

In this paper, we mixed an LCC-SiCN composite slurry that enables the fabrication of high-temperature thin-film temperature sensors by screen printing in a simple and fast way. Scanning electron microscopy (SEM) was employed to observe the surface morphology of the sensitive layer. The crystalline phases before and after annealing were analyzed using X-ray diffraction. Additionally, the elemental distribution of the annealed films was examined through Energy Dispersive Spectroscopy (EDS). Finally, X-ray photoelectron spectroscopy (XPS) was utilized to characterize the presence of relevant functional groups. The results of performance testing indicate that, compared to other sensor devices with unprotected SiCN composite fillers, the sensor demonstrated exceptional high-temperature oxidation resistance, stability, and conductivity following air annealing at 1200 °C. The sensor was capable of measuring temperatures up to 1100 °C and maintaining stability at 1100 °C for a duration of 1 h, with a minimal resistance change of only 3%. Moreover, the sensor exhibited remarkable sensitivity at 200 °C, showcasing a TCR close to −7900 ppm/°C.

## 2. Experimental Method

### 2.1. Fabrication of the Thin-Film Sensor

La(Ca)CrO_3_ powder, with an average particle size of 3 μm, was a calcium-doped LaCrO_3_ with better electrical conductivity than LaCrO_3_. PSN2, a precursor solution for SiCN ceramics, was purchased from the Institute of Chemistry, Chinese Academy of Sciences (Beijing, China). The fabrication process of the thin film sensor was divided into three main steps that are shown schematically in Figure 1. The screen printable high-temperature resistant slurry was obtained by mixing La(Ca)CrO_3_ powder and PSN2 in a container and then stirring for 2 h with a magnetic stirrer. Next, a custom mask was applied to an alumina substrate with platinum leads and the sensitive grid was printed by screen printing. Finally, the sensor was subjected to holding in air at 1200 °C for 1 h using a GSL1700 high-temperature tube furnace, followed by cooling to room temperature. The furnace is manufactured by Hefei Kejing Material Technology Co. (Hefei, China). This process fabricated the sensor that had undergone annealing treatment in the air. The finished sample is shown in Figure 2.

### 2.2. High-Temperature Test System and Methods

The high-temperature test system consisted of a quartz tube furnace (OTF-1200X, MTI KJ GROUP, Hefei, China), a data acquisition instrument (KEYSIGHT 34972A), a K-type thermocouple (KPS-IN600-K), and a computer as shown in Figure 3a. KEYSIGHT 34972A manufactured by Keysight Technologies, Inc. (Santa Rosa, CA, USA). KPS-IN600-K manufactured by Xinghua Suma Electrical Instrument Co. (Xinghua, China). The quartz tube furnace allowed for temperature range control from room temperature to 1200 °C. The data acquisition device records resistance changes and K-type thermocouple data. The hot end of the thermocouple and the thin film sensor were placed in the same temperature zone of the tube furnace. In this way, the temperature recorded by the thermocouple could be approximated as the real-time temperature at the location of the thin-film temperature sensor. Due to the slow cooling rate of the tube furnace, the sensor was tested from 200 °C to 1100 °C in an air environment instead of from room temperature to 1100 °C. In addition, as shown in Figure 3b, the four-wire method was employed to measure resistance, mitigating the impact of electrode and wire resistance.

### 2.3. Characterization Methods

Scanning electron microscopy (SEM) images and energy dispersive spectroscopy (EDS) were obtained using Zeiss Sigma 300 (Jena, Germany). Raman spectra were measured by Lab RAM HR Evolution (Beijing, China). X-ray photoelectron spectroscopy (XPS) was measured using Thermo Scientific ESCALAB Xi+ (Waltham, MA, USA). X-ray diffraction (XRD) was measured by Shimadzu’s XRD-6100 (Kyoto City, Japan).

## 3. Results and Discussion

### 3.1. Characterization Analysis of LCC-SiCN

The detailed morphology of the as-synthesized LCC-SiCN composite thin film was first investigated. Figure 4 shows the SEM image of the LCC-SiCN thin film before annealing. It was found that before annealing, the LCC particles were uniformly dispersed in the PDC, which bound the particles together and acted as a binding and curing agent. Figure 5a shows the surface morphology of the LCC-SiCN thin film after annealing at 1200 °C for one hour in the air. Compared to before annealing, the SiCN precursor shrunk after annealing in the air, along with the overflow of small molecules and the oxidation of carbon elements. The literature also reported that the pyrolysis of polymer-derived SiCN under air generates a SiO_2_ glass phase [33], which is amorphous and can adhere the LCC particles well to the substrate. Meanwhile, it can be seen that the LCC particles were no longer isolated, and the particles were interlaced with each other. The film was free of voids and cracks, forming a continuum and increasing the interparticle contact surface. This facilitated the migration of carriers between LCC particles, resulting in a well-conducting channel. The crystalline phases before and after annealing were analyzed by X-ray diffraction. Figure 6 shows the characteristic peaks of La(Ca)CrO_3_ before and after annealing (with a very close XRD pattern to LaCrO_3_). Due to the amorphous state of the generated SiO_2_, no new peaks were observed in the XRD.

Further, as shown in Figure 5b–f, the elemental distribution of the annealed films was analyzed by EDS, which showed that the LCC-SiCN composite was mainly composed of O, Si, Ca, Cr, and La with a uniform distribution. These results indicate that the SiCN precursor is a carrier to bind the LCC particles, and that the LCC particles have P-type semiconductor properties with potent antioxidant properties. After air annealing, the reaction between SiCN precursor and air caused shrinkage, which increased the contact surface between particles and enhances the electrical conductivity of PDC/LCC. Therefore, the LCC-SiCN composites had outstanding high-temperature stability performance, oxidation resistance, and electrical conductivity.

Figure 7a illustrates the presence of the Si-O peak at 102.56 eV and the Si-N peak at 101.41 eV. Figure 7b indicates the appearance of the Si-N peak at 397.75 eV, the C=N peak at 399.21 eV, and the C-N peak at 400.35 eV. Figure 7c shows the occurrence of the La^3+^ 3d_5/2_ peak at 835.21 eV, the La^3+^ 3d_3/2_ peak at 851.81 eV, the La^3+^ 3d_5/2_ peak at 838.96 eV, and the La^3+^ 3d_3/2_ peak at 855.69 eV. Figure 7d presents the presence of the Cr^3+^ 2p_3/2_ peak at 576.99 eV, the Cr^3+^ 2p_1/2_ peak at 586.65 eV, the Cr^6+^ 2p_3/2_ peak at 579.08 eV, and the Cr^6+^ 2p_1/2_ peak at 588.99 eV. Figure 7e highlights the appearance of the Ca^2+^ 2p_3/2_ peak at 347.17 eV and the Ca^2+^ 2p_1/2_ peak at 350.73 eV. Figure 7f describes the presence of the C-C peak at 284.8 eV, the C-N peak at 285.87 eV, and the C=N peak at 288.28 eV. From Figure 7a, the presence of Si-O peak indicates that the SiCN ceramic transforms into SiCNO ceramic after annealing, while the conductive material La(Ca)CrO_3_ remains unchanged.

### 3.2. High-Temperature Electrical Properties

To illustrate the future application of LCC-SiCN thin-film temperature sensors in high-temperature measurement, heating and cooling temperature cycle tests were performed to verify the sensor repeatability and high-temperature oxidation resistance. Figure 8 shows the resistance-temperature relationship curve after three heating and cooling cycles. The LCC-SiCN thin film temperature sensor had high repeatability and superior oxidation resistance in the temperature range of 200~1100 °C. The resistance monotonically decreased as the temperature increased, and the temperature versus resistance curve follows a nonlinear relationship, showing typical NTC characteristics and further testing the step response of a sensor. Setting the temperature from 200 °C to 500 °C, we held the temperature for half an hour and then held it every 200 °C for half an hour until it reached 1100 °C and then set the cooling step symmetrically. In this way, we could obtain the curve shown in Figure 9. It can be seen that each temperature step corresponds to the sensor resistance value, the sensor resistance value change, and the temperature step change trend of the synchronization. The resistance values corresponding to the temperature rise and temperature fall of the temperature steps have a high degree of consistency.

In addition, durability and stability at high temperatures are critical for high-temperature sensors. Therefore, the sensor was tested to keep the resistance change in the air environment at 1100 °C for one hour. Figure 10 shows a slow increase in the resistance of the sensor within one hour with an overall change of nearly 3%, which is a substantial improvement in oxidation resistance relative to the currently reported PDC composite high-temperature sensors [5,9,10]. The migration of oxygen ions occurs through oxygen vacancies in the lattice, and the rate is influenced by factors such as temperature and oxygen partial pressure. Apart from oxygen ion conduction, La(Ca)CrO_3_ also possesses electron conductivity. The transition metal element Cr facilitates electron transfer between various oxidation states, enabling the material to conduct electricity. However, at high temperatures, the continuous volatilization of Cr leads to an increase in oxygen vacancies, consequently causing a decline in the oxygen ion conduction phase and reduced electron conduction. As a result, the overall conductivity diminishes, and resistance increases. In summary, the aforementioned factors contribute to the observed 3% change in resistance of the sensor after being heated at 1100 °C for one hour.

Figure 11 shows the relationship between the natural logarithm of the resistance (ln R) and the absolute temperature (1000/T) reciprocal for the LCC-SiCN composite thin film temperature sensor. Using the first-order linear equation fit, as shown in the figure by the straight blue line, had a goodness of fit as close to 0.992. The result indicates a linear correlation between lnR and 1000/T, which shows that the electronic conductivity mechanism of LCC-SiCN composites follows the small polarization conduction mechanism of thermal excitation. It also indicates that the LCC-SiCN composites mainly reflect the conductivity properties of LCC [34,35].

These phenomena are also found in other perovskites NTCs as well [36]. This mechanism is generally expressed as Arrhenius relationship:(1)$$R=R_{0}e^{\beta (\frac{1}{T})}$$
(2)$$\frac{1}{T}=A+B\ln(R)+C(\ln(R){)}^{3}$$
(3)$$\mathrm{TCR}=\frac{R2-R1}{R1*(T2-T1)}$$
where R is the resistance at temperature T (°C), A, B, and C are the coefficients of SHHE, R_0_ is the resistance at the reference temperature T_0_, and β is the material constant. R1 is the resistance value when the temperature is T1; R2 is the resistance at temperature T2.

The Steinhart–Hart equation (SHHE) [37] (as shown in Equation (2)) is an empirical formula that has been determined to be the best mathematical expression for the resistance-temperature relationship in NTC thermistors and NTC probe assemblies [36]. It is also a third-order polynomial that better describes the nonlinear behavior of the resistance-temperature relationship. The red line in Figure 11 was fitted using the SHHE, and the goodness-of-fit is close to 0.998, reflecting the high goodness-of-fit over a wide range of 200 to 1100 °C. Furthermore, the average temperature coefficient of resistance [38] (TCR) is defined by Equation (3). Figure 12 shows that the variation of the TCR of the temperature sensor in the range of 200 to 1100 °C is close to −7900 ppm/°C at 200 °C. As the temperature increased, the absolute value of TCR gradually decreased, and its TCR is close to −1000 ppm/°C at 1100 °C. The sensitivity of LCC-SiCN to the change of temperature is nearly two times better than the reported [9,10] TCR of PDC high-temperature sensors.

## 4. Conclusions

In summary, we propose the application of an innovative LCC-SiCN slurry by screen printing to develop ultra-high temperature thin film sensors. The simple process and inexpensive materials make the LCC-SiCN sensor promising as an economical high-temperature sensor. The SiCN precursor in the composite material acts as a carrier to disperse the LCC particles. After high-temperature annealing, the SiCN precursor produces SiO_2_ in an amorphous state that bonds and anchors the LCC particles to the substrate. Therefore, the LCC particles play a critical conductive role in the sensor. The resistance temperature profile of the sensor also shows an evident NTC characteristic of typical LCC material in the range of 200 to 1100 °C. Meanwhile, the TCR of the sensor decreases with increasing temperature, varying from −7900 ppm/°C at 200 °C to −1000 ppm/°C at 1100 °C. Remarkably, the sensor possesses excellent anti-oxidation performance with a 3% change in resistance at 1100 °C for one hour of holding time. Therefore, screen printing LCC-SiCN slurry for high-temperature thin film sensors can solve the constraints of traditional physical vapor deposition and high-temperature resistant materials prone to oxidation. This method has great potential for applications in high-temperature sensors.

## Figures and Tables

**Figure 1 micromachines-14-01719-f001:**
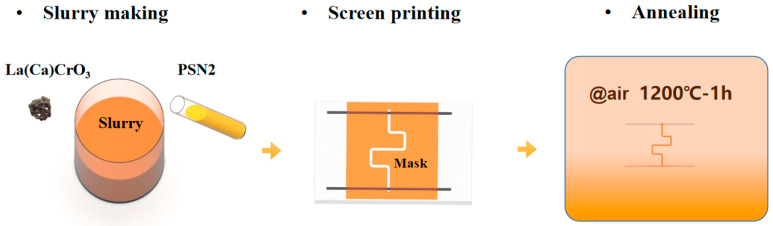
The fabrication process of the LCC-SiCN composite thin-film temperature sensor.

**Figure 2 micromachines-14-01719-f002:**
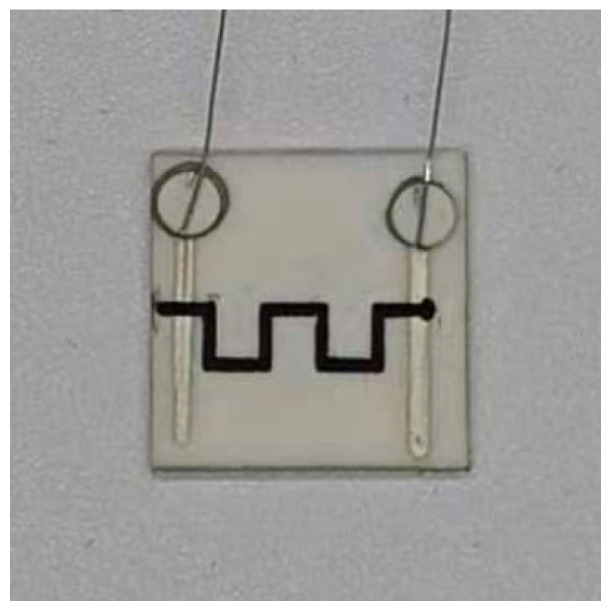
The LCC-SiCN composite thin-film temperature sensor after annealing process.

**Figure 3 micromachines-14-01719-f003:**
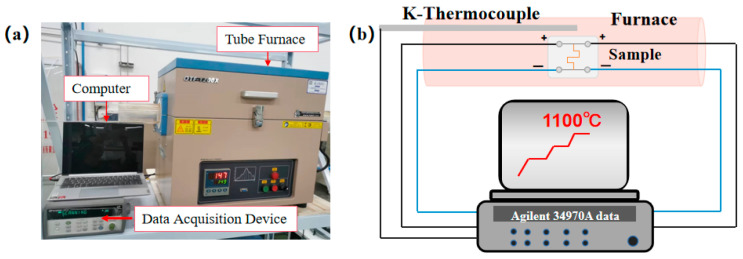
(**a**) High-temperature temperature test system. (**b**) Four-wire configuration test method diagram.

**Figure 4 micromachines-14-01719-f004:**
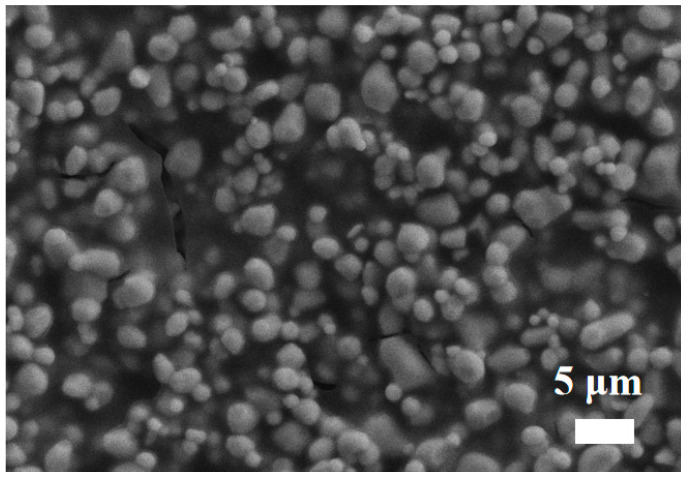
The SEM image of the LCC-SiCN thin film before annealing.

**Figure 5 micromachines-14-01719-f005:**
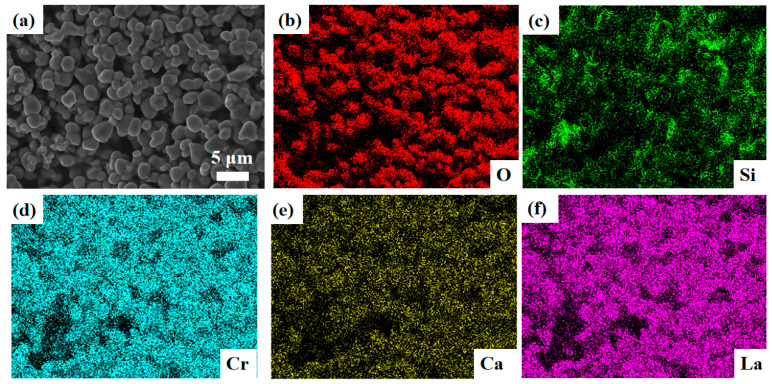
(**a**) The SEM image of the LCC-SiCN thin film after annealing. (**b**) The EDS image of O distribution. (**c**) The EDS image of Si distribution. (**d**) The EDS image of Cr distribution. (**e**) The EDS image of Ca distribution. (**f**) The EDS image of La distribution.

**Figure 6 micromachines-14-01719-f006:**
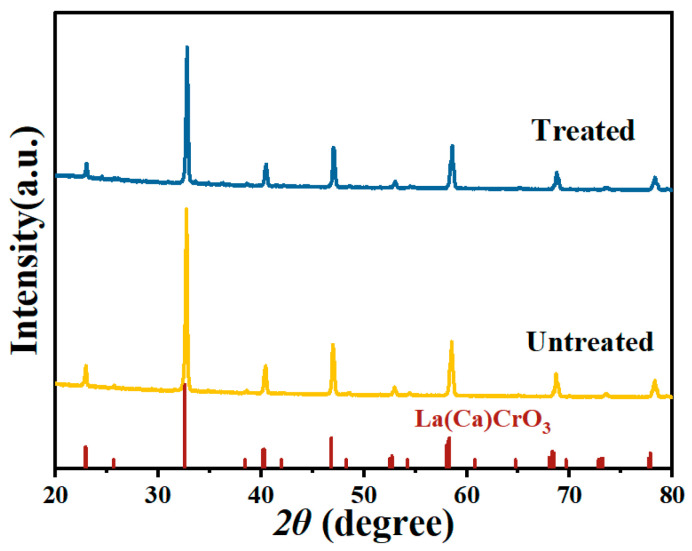
XRD pattern of the sensitive film.

**Figure 7 micromachines-14-01719-f007:**
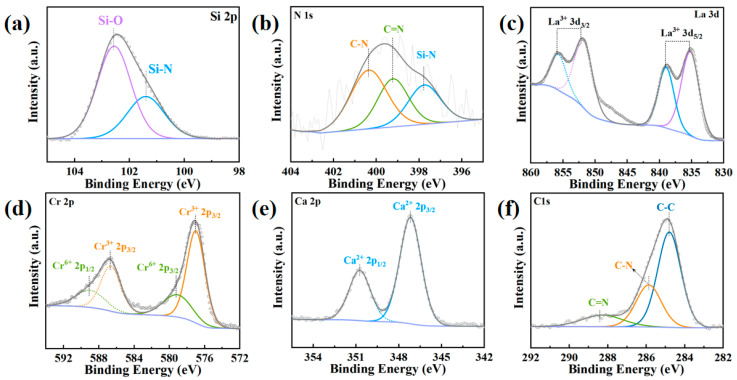
(**a**) Binding energy of Si 2p. (**b**) Binding energy of N 1s. (**c**) Binding energy of La 3d (**d**) Binding energy of Cr 2p (**e**) Binding energy of Ca 2p, (**f**) Binding energy of C ls.

**Figure 8 micromachines-14-01719-f008:**
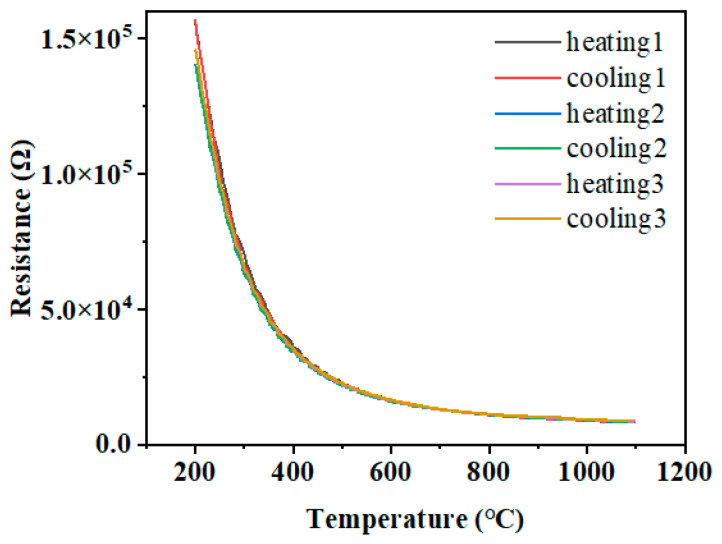
The result of the thermal cycle test.

**Figure 9 micromachines-14-01719-f009:**
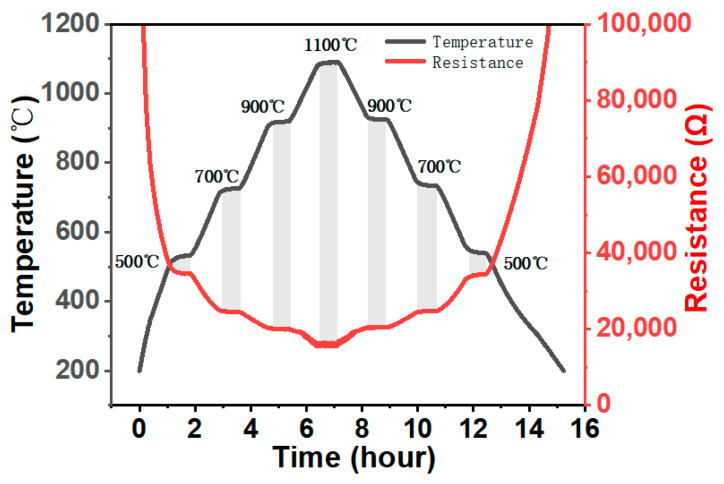
Temperature step response of the sensor.

**Figure 10 micromachines-14-01719-f010:**
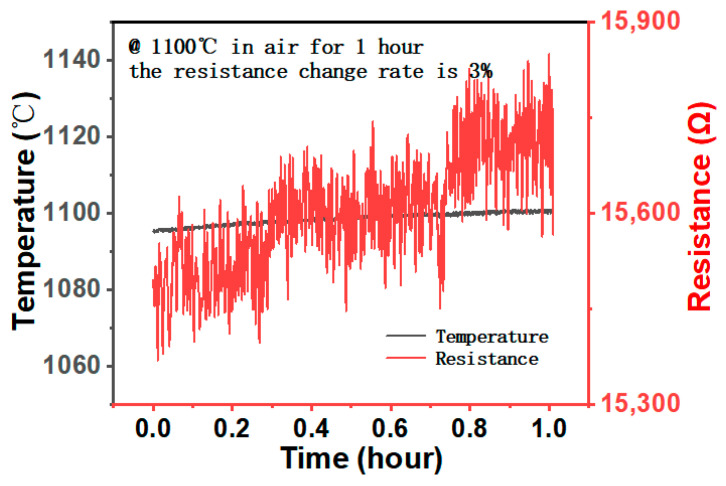
The sensor resistance changes in the air at 1100 °C for 1 h.

**Figure 11 micromachines-14-01719-f011:**
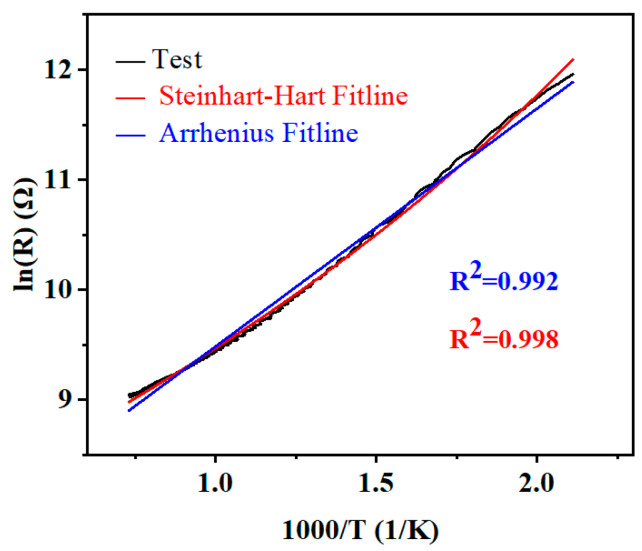
Experimental results of the temperature dependence of the electrical resistance compared with fitting by SHHE.

**Figure 12 micromachines-14-01719-f012:**
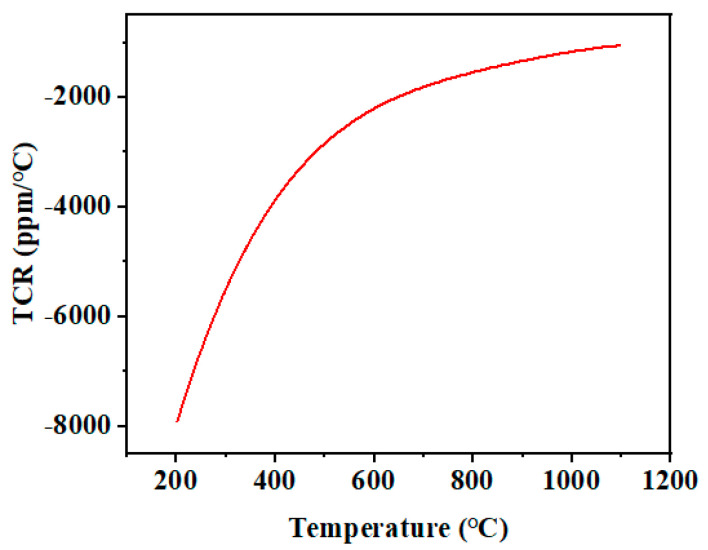
TCR of the sensor.

## Data Availability

Not applicable.

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
