# Peer review of "La(Ca)CrO3-Filled SiCN Precursor Thin Film Temperature Sensor Capable to Measure up to 1100 °C High Temperature"

_micromachines, 2023, doi:10.3390/mi14091719_

Round 1

Reviewer 1 Report

In this manuscript, the authors described their way to fabricate La(Ca)CrO3/SiCN thin film that can potentially be used as high temperature sensors. In most part, the manuscript clearly presents the research motivation and objectives of the authors, described their methodology well, and backed up their conclusion with data of high quality. I have only a few remarks. 

In Part 2.1 of Experimental Method, what is the doping concentration of La(Ca)O3 powder? Did authors synthesize the powder by themselves? If so, what was the process? If the powder was purchased, what was the brand (if available) and purity?

Reviewer 2 Report

This manuscript presents an approach using LCC-SiCN slurry through screen printing to fabricate high temperature thin film sensor. The manuscript outlines the synthesis process, characterization methods, and performance evaluation of the developed sensors. While the work shows potential, I have identified specific areas that require significant revisions and improvements to enhance the manuscript's clarity. Therefore, I recommend a major revision of the manuscript to address these issues before it can be considered for publication.

1. The introduction should provide a comprehensive review of existing works related to LCC, as it is not a new material. It is important to highlight the unique aspects of the current work and how it differs from previous research.

2. Page 4: How can the conclusion that the SiCN precursor is a carrier to bind the LCC particles be drawn solely based on the SEM images of LCC-SiCN? I believe it would also be necessary to include an image of LCC alone to support this assertion.

3. The authors mention on page 6 that the XPS results indicate the transformation of SiCN ceramic to SiCNO ceramic after annealing. Please provide a detailed explanation.

4. While the manuscript focuses on the temperature range of 200-1100°C, I am curious about what could potentially happen if the temperature goes below 200°C or above 1100°C.

5. Page 7: What causes 3% resistance change observed after subjecting the sensor to heating at 1100°C for one hour?

6. For the semiconductor material, I assume the TCR is negative, as plotted in Figure 12. However, why do the authors present a positive TCR number in the text on page 8?

Moderate editing of the English language is required to improve the overall readability and coherence of the manuscript

Round 2

Reviewer 2 Report

I have carefully reviewed the revised manuscript submitted by the authors. The authors have made improvements to the manuscript, providing clarifications and addressing the issues raised. Based on the revisions made, I believe the manuscript is now suitable for publication.

The manuscript would benefit from proofreading for language and grammatical errors.